# Efficacy of Regional Chemotherapy Approach in Peritoneal Metastatic Gastric Cancer

**DOI:** 10.3390/jcm10225322

**Published:** 2021-11-15

**Authors:** Kornelia Aigner, Yogesh Kumar Vashist, Emir Selak, Sabine Gailhofer, Karl Reinhard Aigner

**Affiliations:** Clinic for Surgical Oncology, Medias Klinikum Burghausen, 84489 Burghausen, Germany; kornelia.aigner@medias-klinikum.de (K.A.); e.selak@medias-klinikum.de (E.S.); s.gailhofer@medias-klinikum.de (S.G.); prof-aigner@medias-klinikum.de (K.R.A.)

**Keywords:** regional chemotherapy, isolated abdominal perfusion, intra-arterial infusion, stop-flow perfusion, peritoneal carcinomatosis, gastric cancer

## Abstract

Peritoneal spread is frequent in gastric cancer (GC) and a palliative condition. After failure to systemic chemotherapy (sCTx) remaining therapeutic options are very limited. We evaluated the feasibility and efficacy of locoregional chemotherapy (RegCTx) in peritoneal metastatic GC. In total, 38 (23 male and 15 female) patients with peritoneal metastatic GC after failure of previous sCTx and unresectable disease were enrolled in this study. Using the hypoxic abdominal stop-flow perfusion, upper abdominal perfusion and intraarterial infusion technique in total 114 cycles with Cisplatin, Adriamycin and Mitomycin C were applied. No significant procedure related toxicity was noticed- especially no Grade 3 or 4 toxicity occurred. With the RegCTx approach a median overall survival of 17.4 months was achieved. Patients who had undergone previously resection of the GC the median overall survival was even better with 23.5 months. RegCTx is a promising, safe and efficient approach in diffuse metastatic GC. The evaluation of RegCTx in the setting of multimodal treatment approach at less advanced stages is also warranted.

## 1. Introduction

Gastric cancer (GC) is the fifth most common malignancy worldwide and peritoneal spread is often already present at the time of first diagnosis or does occur as metastatic site in majority of the cases, resulting in a palliative condition with unsatisfactory outcome and an overall survival (OS) of less than six months with best supportive care only [1,2,3]. In recent times, many systemic chemotherapy (sCTx) regimes have emerged with promising results [1]. In addition, surgery is also not excluded in metastatic GC nowadays. Hyperthermic intraperitoneal chemotherapy (HIPEC), pressurized intraperitoneal aerosol chemotherapy (PIPAC) and repeated intraperitoneal chemotherapy (RIPEC) are also evaluated in the management of peritoneal carcinomatosis in GC within the multidisciplinary tumour board discussions [3,4,5,6,7,8,9].

However, the patient is at a dead end if the relapse is diffuse and re-surgery not possible or after failure to sCTx and or if the overall general condition is too poor to perform a tumour specific therapy [1]. In addition, second line sCTX are critically discussed in metastatic GC and are very limited. Apart from the oncological aspect the quality of life (QoL) is the most important factor in management of metastatic GC patients [10].

Regional chemotherapy (RegCTx) is an oncological approach with very low toxicity profile and high tumour response due to high cytotoxic drug concentration in the isolated perfusion bed [11,12,13,14,15,16,17,18,19]. In addition, the therapy can be focused on limited regions if necessary, using the same technique e.g., hypoxic abdominal stop-flow perfusion (HAP), upper abdominal perfusion (UAP), isolated thoracic perfusion (ITP) and hypoxic pelvic perfusions (HPP) as well as intra-arterial infusion. The restricted perfusion bed that is treated during RegCTx allows potentiation of drug concentration levels at the tumour site compared to sCTx although the applied drug dosages are limited to 20–50% of those used in sCTx. Furthermore, the possibility to perform a chemo-filtration ensures lowest systemic toxicity effects. RegCTx efficacy has been proven in many cancers but has not been reported in GC series [11,12,14,15,17,18,19,20]. Here we report on our institutional experience with 38 far advanced metastatic GC patients undergoing RegCTx after failure to sCTx.

## 2. Patients and Methods

### Characterization of the Study Population

In total, 39 patients underwent RegCTx at our institution. One patient was lost to follow up and was therefore excluded from this analysis. Complete data and follow up was available for total 38 patients. All patients had histologically proven GC. None of the patient had received RegCTx before. The median age of the study population was 54 years ranging from 37 to 74. There were 23 (60%) males and 15 (40%) females. Majority of the patients (*N* = 22) had severely limited general condition with Karnofsky index ≤ 60%, and in addition, 18 (47%) patients had ubiquitous ascites. Peritoneal metastases were present in all patients and 26 (68%) had at least additional single or multiple other site metastases. Furthermore 18 (47%) had undergone a resectional procedure for the GC and 19 (50%) patients had previously failed sCTx—mostly platinum based. In addition, 10 (26%) patients had more than only first line sCTx attempt. Four (10%) had undergone radiotherapy but 12 (32%) were completely therapy naïve before RegCTx. Table 1 depicts all characteristics of the study population.

## 3. Cytotoxic Drugs and Methods

### 3.1. Regional Chemotherapy Techniques

Hypoxic abdominal stop-flow perfusion (HAP) [14] is a technique that allows an isolated perfusion of the abdominal region. Perfusion balloon catheters are placed in the vena cava inferior and the aorta, both right beneath the diaphragm and pneumatic cuffs around the thighs—just below the inguinal region—the peritoneal region is connected to an extracorporeal circuit. The catheters are inserted through an incision in the groin area via the femoral artery and vein. After an intra-arterial bolus infusion into the aorta both balloons are blocked and a stopped blood flow phase with very high drug concentrations in the abdomen arterial tree is created for five minutes. Afterwards, the isolated perfusion is run for five minutes. It generates highly concentrated drug levels in the isolated region under hypoxic conditions. Another five minutes of perfusion with inflated balloons is conducted with contemporary chemo-filtration. After deflating the balloons chemo-filtration is continued until five liters of substitutional volume is reached. Figure 1a,b demonstrates the principle of HAP and the structure of the used balloon catheter at our institute. Drugs are chosen according to their cytotoxic potential under hypoxic conditions. HAP was performed 42 times in our patient cohort.

If cross sectional imaging demonstrated high tumour burden (e.g., recurrence in the primary tumour region), the HAP was focused to the upper abdominal region by combination with a prior upper abdominal perfusion (UAP) that was immediately followed by the above-described HAP.

For the UAP, in the first step, the venous balloon is positioned right beneath the diaphragm and the arterial balloon is placed right beneath the celiac trunk. Chemotherapeutic drugs are infused during one minute from the tip of the arterial balloon ensuring the entire cytotoxic drug is spread through the celiac perfusion bed. Parallel to this the venous balloon is inflated. Thereafter the arterial balloon is immediately slipped upstream and positioned above the celiac trunk. A stopped blood flow phase with very high drug concentrations in the upper abdomen results from this replacement of the arterial catheter and lasts for the first five minutes.

For Step Two, the perfusion is run through holes in the catheter tubes downstream the arterial and venous balloons. This resulted in still relatively high drug concentrations in the whole abdominal perfusion bed under hypoxic condition for five minutes. The second step is identical to the HAP treatment and is also followed by chemo-filtration as described above. UAP and HAP were performed 40 times in our patient cohort, and all were followed by chemo-filtration.

In patients who were not suitable for general anaesthesia, or a potential strong tumour necrosis post perfusion was feared with sepsis, an intra-arterial infusion [13] was applied only. For this, an angiographic sidewinder catheter is inserted via the femoral artery into the celiac trunk. Drugs are infused as short infusions during five to 12 min with short term plateau of considerably high drug concentrations in the perfusion bed. Twenty-four treatment cycles have been applied with intra-arterial infusion, thereof seven were performed with an angiographic catheter under full anaesthesia and followed by chemo-filtration, 14 have been applied with an angiographic catheter without full anaesthesia and chemo-filtration and three have been applied via an implanted arterial port catheter only.

For one patient with lung metastases, an isolated thoracic perfusion (ITP) was performed twice. This technique is conducted with the same balloon catheters as for UAP and HAP, but the isolated circuit is located above the balloons in the thoracic region. Pneumatic cuffs around the upper forearms reduce the perfusion bed volume and ensure high drug concentrations in the isolated perfusion circuit.

For three patients with metastases to the ovaries, six hypoxic pelvic perfusions (HPP) were performed respectively. This technique is conducted in the same manner and catheters as for UAP, HAP and ITP but the venous and arterial balloons are placed right above the iliac bifurcation and the isolated circuit is further restricted to the pelvic region by pneumatic cuffs around both thighs.

### 3.2. Cytotoxic Drugs

For the treatment under hypoxic condition, such as abdominal perfusion and pelvic perfusion, Cisplatin, Adriamycin, and Mitomycin C are used as they have equal (Cisplatin) or enhanced (Adriamycin and Mitomycin C) cytotoxic potential under anaerobic conditions. Experimental in-vitro cell culture studies have demonstrated that Mitomycin C and doxorubicin have increased cell toxicity under hypoxic conditions and Cisplatin has equal cell toxicity under aerobic and hypoxic conditions [21]. Patients who have reached the cumulative threshold for these drugs or who do not or no longer respond to even higher concentrated local treatment, a switch was performed to a combination with either Mitoxantron, Paclitaxel, Gemcitabine, Docetaxel, or 5-FU. Twenty-five patients required a change of drug combination—mostly for the later cycles.

Drug dosages for perfusions were 50–60 mg Cisplatin, 30–40 mg Adriamycin, and 10–20 mg Mitomycin c, respectively. Intra-arterial infusions have been conducted with 30–40 mg Cisplatin, 10–30 mg Adriamycin, and 10–20 mg Mitomycin. Drugs were allotted higher to infusions that have been followed by chemo-filtration compared to infusions without chemo-filtrations.

### 3.3. Treatment Cycles

Regional chemotherapy has been applied in treatment cycles. Each treatment cycle consisted of either one isolated perfusion or one intra-arterial treatment followed by chemo-filtration or intra-arterial infusion with the total cycle dosage distributed to four sequential days and no chemo-filtration. Each therapy cycle was followed by a three-week therapy free interval.

Fifteen patients had ≥5 cycles and 23 patients received up to 4 treatment cycles. The techniques were alternated for different cycles for each patient if different metastatic locations were to be treated.

#### Statistical Analysis

Kaplan–Meier estimations have been calculated with MediasStat software version 28.5.14.

## 4. Results

### 4.1. Adverse Events

Incidence of general side effects such as nausea and fatigue have been very low and only mild and did not require any additional medication to standard to post RegCTx protocol. Toxicity Grades 3 and 4 never occurred. Lymph fistula at the inguinal incisional site was the most frequent adverse event and occurred in about 30% of the cases. All cases were managed conventionally and no re-operation for the fistula was necessary. Hair loss, hand-foot syndrome, and neuropathy never occurred. 

### 4.2. Response

Responses to the treatment have been measured under RECIST criteria. Usually, after two cycles of therapy, a CT scan has been conducted. For 17 patient sequential results were reported in 22 scans. Partial response (PR) and stable disease (SD) have been observed in 45% and 27% of the scans. Progressive disease (PD) has been observed in 6 scans. Verified response according to the treatment cycles showed a benefit for the first four cycles of therapy with a PR of 30%, 40% and 30% for treatment cycle number 2, 3, and 4, respectively. Stable disease has been observed in 33% and 67% for treatment cycle number 2 and 3, respectively. Progressive disease has been observed in 0%, 33%, 17%, 33%, and 17%, for treatment cycle 2, 3, 4, 5 and 6, respectively. Partial response and SD have not been observed after the 5th cycle.

### 4.3. Survival

For the observed patients, the median OS was 17.4 months. Survival rates reached 73%, 27%, 8.1%, and 5.4% at Years 1, 2, 3, and 4 respectively and remained at 5.4% at year six. Patients who had undergone surgical resection prior to RegCTx had a median OS of 23.5 months. The median OS from the timepoint of RegCTx initiation was 6.7 months with 17.3% (*N* = 7) patients even alive after one year. Interestingly, the previous systemic treatment did not have much impact on the survival time after RegCTx. Even patients with already unresectable peritoneal disease and additional other site metastases at the time of first diagnosis, had still a median of eight months survival after RegCTx compared to six months for patients with diagnosis at an early stage and surgical resection for example. Similar results have been observed for patients regarding the systemic pre-treatment. Patients who have undergone sCTx prior to RegCTx had a median OS of 6 months after RegCTx compared to patients without systemic pre-treatment with 8 months survival after RegCTx. In contrast to this, the OS for patients with or without systemic pre-treatment was 20 months and 13.5 months, respectively. Figure 2 demonstrates the survival cure of the entire cohort.

## 5. Discussion

Peritoneal spread—a palliative condition—will ultimately appear in the majority of GC cases. In recent years, many advances have been made in the field of sCTx as well as applying multimodal treatment regimes [1,3,8,10]. Hyperthermic intraperitoneal chemotherapy (HIPEC), pressurized intraperitoneal aerosol chemotherapy (PIPAC) and repeated intraperitoneal chemotherapy (RIPEC) in the management of peritoneal carcinomatosis in GC have emerged [4,5,6,7,9,22,23]. However, the overall outcome remains unsatisfactory with survival rates between 6 to 18 months only. In addition, severe systemic toxicity and surgery related morbidity and mortality remain unsolved issues and most treatment options are only applicable to patients with overall good general condition (ECOG 0–1) and limited peritoneal spread hence low tumour load [1,3,8,10].

Here we report our experience with RegCTx on a cohort of 38 GC patients with diffuse peritoneal spread and high tumour load with 14 (37%) patients having additional multiple metastatic site different from peritoneum. Nineteen (50%) of the 38 patients presented with a progress after having undergone sCTx. The median OS in our cohort was 17.4 months with RegCTx.

The Cochrane meta-analysis by Wagner et al. has demonstrated an OS benefit of 6.7 months for sCTx in stage IV GC patients compared to best supportive care. The benefit of combination sCTx was calculated to one month only in that analysis, however it was outlined that the benefit was associated with more systemic toxicity [1,24]. In contrast, our cohort did not experience any Grade 3 or 4 toxicity or required re-hospitalization for RegCTx associated toxicity. Furthermore, the majority of the patients had an initial Karnofsky index of ≤60 translating into ECOG 2 to 4 at the time of RegCTx initiation with most patients having three and few also four, indicating that sCTx was not even possible or applicable in these patients. In addition, the median OS of 17.4 months in our group outlines the efficacy and feasibility of RegCTx in this advanced and highly compromised GC patients.

At present in Germany for HER-2 negative tumours a combination sCTx is recommended [1]. The different available randomized clinical trials from around the world comparing various combination of Cisplatin, 5-FU, S-1, docetaxel and epirubicin have only demonstrated a median OS of less than one year with Grade 3 and 4 toxicity profiles in at least >50% but also reaching up to 82% of the patients [24]. In recent years, the FLOT regime has gained much popularity in the management of advanced GC patients with a proposed lower systemic toxicity profile and median OS of 11.3 months in unresectable metastatic GC cohort. However, only 50% were considered as responder in that trial and the toxicity was much higher in older patients—with Grade 3 and 4 side effects reaching up to 82% [8]. In addition, patients ≥70 years of age did not benefit from the triple combination. Hence, this also indicates towards the justified approach in our cohort with a significant better median OS and no Grade 3 or 4 toxicities—especially taking into consideration that our cohort was not sCTx naïve.

Data for HER2-positive tumours from the ToGA trial imply a benefit for Trastuzumab addition to the systemic component of 5 months with median OS of 16 months versus 11 months without. However, only 22% of the almost 4000 patients enrolled were pathologically eligible for Transtuzumab treatment [25]. In addition, compared to this approach our median OS was slightly better and taken into account that 50% of our patients had already undergone a systemic treatment a direct comparison to the ToGA trial outcome is not justified. In addition, in our GC cohort all patients had a Karnofsky index of ≤60—equivalent to ECOG ≥ 2.

Another aspect to be considered in the management of these highly compromised patients is the duration of therapy. Systemic combinations are recommended for up to 24 weeks and in HER2-positive patients a maintenance therapy is even recommended [1,3].

In contrast, our therapy approach has required a hospital stay for RegCTx of only a few days per cycle and 60% of our patients received one to four cycles. Our data demonstrate that the first three cycles are the most efficient cycles with the best documented disease response. In our cohort patients were able to move around on the day of RegCTx and no documented side effect appeared that required re-hospitalization. Patients could be discharged after each cycle within three to five days from the hospital followed by a three-week therapy free interval. Since quality of life is a major aspect in the management of these patients our approach ensured no impairment in quality of life but a significant improvement. Taken into consideration that we have treated mostly ECOG 2 and 3 and even few ECOG 4 patients, this pinpoints towards justification of this approach without even taking the OS benefit and low toxicity profile into account.

It has to be outlined that most of our patients were not sCTx naive but had undergone more than one sCTx and did present with a progress of the disease. Hence, also in terms of the oncological outcome our data indicate towards a beneficial role of RegCTx in metastatic GC with a median OS of 17.4 months.

The role of second line sCTx in metastatic GC is critically considered. The available studies so far including docetaxel, irinotecan and or Ramucirumab reported a survival benefit of less than five months compared to best supportive care only. This also pinpoints towards the efficacy of RegCTx in metastatic GC [1,3].

HIPEC and PIPAC have been advocated in the last few years in the management of metastatic GC patients [4,5,6,7]. However, the overall data are scarce. The GYMSSA trial reported a median OS of 11 months, and the PERISCOPE II trial is currently recruiting [26,27]. For PIPAC randomized controlled trials in GC are not yet available. The study by Struller et al.—open-label, single-arm, monocentric Phase II trial—showed a median OS of only 6.7 months for the PIPAC in combination with sCTx [28]. However, HIPEC and PIPAC are both now incorporated into clinical management of advanced metastatic GC as many trials are on the way to evaluate the efficacy of these approaches in combination with sCTx. In our patient cohort, the survival of patients who had undergone gastric resection previously even reached median 23.5 months. However, our group of patients was not a candidate for HIPEC since many had even more than one other metastatic site than peritoneum only and also the extent of metastatic disease with the overall poor general condition did not allow a HIPEC or PIPAC approach in these patients. 

RegCTx also allows the treatment of distant metastases as the techniques can be adapted to other regions such as the chest or pelvis and induce higher drug concentrations at the metastatic locations than sCTx enables.

We have been able to demonstrate the safety and efficacy of RegCTx in advanced metastatic GC. However, our cohort is heterogenous and represents patients who would have been only addressed to best supportive care otherwise. Hence, comparable data are missing. Possible subgroups analysis in patients with and without sCTX especially stratified according to specific sCTX drug was not possible but for the sake of complete and transparent data we have addressed this issue in our analysis. Hence, though few numbers may appear not rationale at first sight it is likely to be due to small subgroups. However, the overall benefit with a median OS of 17.4 months remains indisputable. Direct comparison to other forms of cancer therapy is certainly only possible to a limited extent as well but the characterization and outcome of our patient cohort indicates the clinical dilemma of impossibility to undertake a tumour specific therapy in these patients in general on the one hand but also the possibility of RegCTx as a tumour specific approach with very low toxicity profile on the other hand.

In conclusion, RegCTx offers a low toxicity associated therapy approach with a survival benefit over current recommended sCTx options in far metastatic advanced GC. Future studies are required to evaluate the role of RegCTx in the multimodal management of GC also in less advanced disease stages especially.

## Figures and Tables

**Figure 1 jcm-10-05322-f001:**
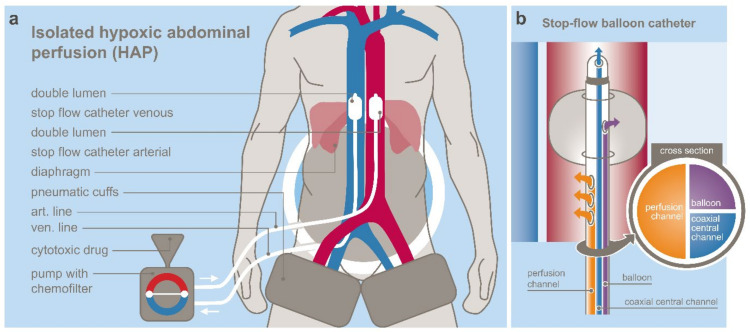
(**a**) schematic principle of hypoxic abdominal perfusion; (**b**) stop-flow catheter design.

**Figure 2 jcm-10-05322-f002:**
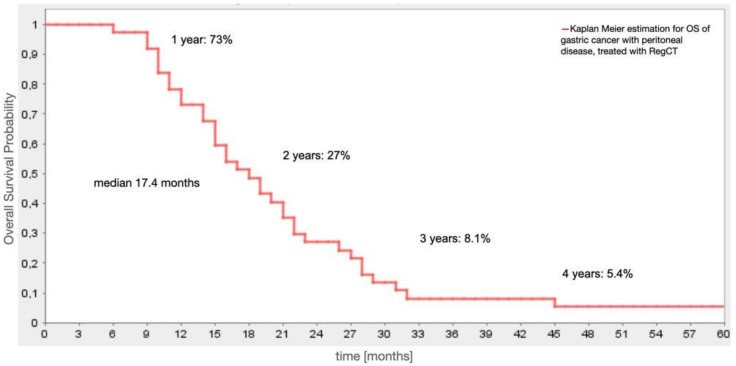
Overall survival of the entire patient cohort (*N* = 38).

**Table 1 jcm-10-05322-t001:** Patient characteristics.

Value		*N* (%)
Total		38 (100)
Gender	male	23 (60)
	female	15 (40)
Age—median (range)		54 (37–74)
Karnofsky Index	30–60%	24 (63)
	70–100%	14 (37)
Ascites	2 quadrants	4 (11)
	4 quadrants	18 (47)
Metastases Spectrum	peritoneal	38
	multiple sites (other than peritoneal)	14 (37)
	single site (other than peritoneal)	12 (32)
	peritoneal only	12 (32)

	liver	11 (29)
	lymphatic	13 (34)
	lung	1 (2)
	bone	2 (5)
	other	10 (26)
Pre-treatment	systemic chemotherapy	19 (50)
	surgery	18 (47)
	no pre-treatment	12 (32)
	radiotherapy	4 (10)
Chemotherapy Spectrum	first line chemotherapy	9 (24)
	multiple lines chemotherapy	10 (26)
	platin-based chemotherapy	18 (47)
	others	15 (39)

## Data Availability

The data presented in this study are available on request from the corresponding author. The data are not publicly available due to data protection regulations and ethical issues.

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
