# Peer review of "Efficacy of Regional Chemotherapy Approach in Peritoneal Metastatic Gastric Cancer"

_jcm, 2021, doi:10.3390/jcm10225322_

Round 1

Reviewer 1 Report

Overall, the manuscript reports an interesting and promising potential approach for treatment of advanced metastatic GC patients, especially with impaired overall condition. The manuscript is very well written and easy to follow.

I would ask the author to address one single aspect: the authors report OS of 17.4 months. However, this overall survival is the endpoint after multiple treatments including Reg CTx and not the benefit of RegCTx only. Therefore, I would ask the authors to clearly outline the respective survival benefit after RegCTx (information  partly has already been provided but is not conclusive in its present form) in addition to the already applied treatment and to compare this benefit to best supportive care or second line sCTx treatment according to the literature. In this context it might make sense to provide data only for major subgroups of patients such as initially resected in curative intend but than relapsed versus initially metastatic status with sCTx treatment before RegCTx versus initially metastatic status without treatment before RegCTx.

Reviewer 2 Report

Dear Authors,

I would like to congratulate you on these results.

Just few minor issues:

1. Are there any changes in the serum biochemistry during or after the regional chemotherapy  - especially in inflammatory markers?

 2. How does tumor marker respond to the treatment.?

Reviewer 3 Report

This manuscript has been described the efficacy and the safety of hypoxic abdominal stop-flow perfusion, upper abdominal perfusion and intraarterial infusion technique for peritoneal metastatic GC. It is impressive this treatment available even for patients with ECOG2 or 3. 

  1. Author called HAP, UAP, HPP and intra-arterial infusion as regional chemotherapy. However, intraperitoneal chemotherapy (IPEC) also should be included regional chemotherapy. Therefore, article title should be changed not to confuse readers, because IPC is one of the popular treatments for peritoneal metastatic GC.

  1. In Introduction, author described HIPEC and PIPAC were another treatment option for peritoneal metastatic GC. However repeated IPEC (RIPEC) with taxanes is also one of treatment option with powerful potential. Author should mention this point (ref.: Kitayama J et al. Treatment of peritoneal metastases from gastric cancer. Ann Gastroenterol Surg 2018). Also, please mention in Discussion.

  1. In Patient and Methods, author should clarify the degree of peritoneal metastasis of each patient according to peritoneal carcinomatosis index (PCI) or Japanese classification of gastric carcinoma, because treatment outcome depends on the degree of peritoneal metastasis.

  1. In Cytotoxic drug and methods, author described Cisplatin, Adriamycin and Mitomycin C have been chosen according to their cytotoxic potential under hypoxic condition. Author should describe detail this point such as hypoxia leads acidic environment, or active range of pH of each drug.

  1. Peritoneal metastatic tumor of GC consists rich stroma tissue including fibrosis resulting in poor blood perfusion so called blood-peritoneal barrier (BPB) in different from lymph node metastasis and hepatic metastasis. Author should present difference of drug concentration in peritoneal metastatic tumor between sCTx and so called RegCTx.

  1. In Results, assessment of treatment response according to RECIST is important. However, treatment result of peritoneal metastasis is more important. If impossible, change of ascites volume or negative peritoneal cytology rate should be assessed.

  1. In Discussion, articles concerning cytoreductive surgery and HIPEC or RIPEC must be choses fairly. Below articles could be recommended.

Yonemura Y et al. Long term survival after cytoreductive surgery combined with perioperative chemotherapy in gastric cancer patients with peritoneal metastasis. Cancers. 12 (1), E116: 2020.

Ishigami H et al. Phase III trial comparing intraperitoneal and intravenous paclitaxel plus S-1 versus cisplatin plus S-1 in patients with gastric cancer with peritoneal metastasis: PHOENIX-GC trial. J Clin Oncol 2018.

Round 2

Reviewer 3 Report

Although author did not improve for all reviewer's comments, I hope it will be reflected the next multicenter prospective study.